# Protein Phase Separation during Stress Adaptation and Cellular Memory

**DOI:** 10.3390/cells9051302

**Published:** 2020-05-23

**Authors:** Yasmin Lau, Henry Patrick Oamen, Fabrice Caudron

**Affiliations:** School of Biological and Chemical Sciences, Queen Mary University of London, Mile End Road, London E1 4NS, UK; y.e.y.lau@qmul.ac.uk (Y.L.); h.oamen@qmul.ac.uk (H.P.O.)

**Keywords:** stress, cellular memory, phase separation, prions

## Abstract

Cells need to organise and regulate their biochemical processes both in space and time in order to adapt to their surrounding environment. Spatial organisation of cellular components is facilitated by a complex network of membrane bound organelles. Both the membrane composition and the intra-organellar content of these organelles can be specifically and temporally controlled by imposing gates, much like bouncers controlling entry into night-clubs. In addition, a new level of compartmentalisation has recently emerged as a fundamental principle of cellular organisation, the formation of membrane-less organelles. Many of these structures are dynamic, rapidly condensing or dissolving and are therefore ideally suited to be involved in emergency cellular adaptation to stresses. Remarkably, the same proteins have also the propensity to adopt self-perpetuating assemblies which properties fit the needs to encode cellular memory. Here, we review some of the principles of phase separation and the function of membrane-less organelles focusing particularly on their roles during stress response and cellular memory.

## 1. Introduction

Phase separation is the demixing of a homogeneous mixture in solution to two separated phases, one of which can take the form of an assembled and detectable structure. These assembled structures have been given various nomenclatures including biomolecular super-assemblies, condensates, quinary structures or membraneless organelles, which suggests that a wide array of structures could be included in this classification [1]. One such example of a phase-separated structure is P-granules and is involved in specifying germ cells in *Caenorhabditis elegans* [2,3]. Since phase separation was found to play an important role into the biology of *C. elegans*, a plethora of structures forming through phase separation have been identified and examples of these include nucleolus for ribosome assembly, paraspeckles and nuclear speckles which regulate gene expressions, P-bodies for RNA storage and processing, Cajal bodies for regulation of small nuclear and small nucleolar RNA genes, stress granules for storage of stress-halted proteins and RNA [4,5,6,7,8,9,10,11]. In addition to phase separation being fundamental in cell physiology, its role and perturbation in diseases has emerged as a novel focus to understand the mechanisms of some pathologies such as amyotrophic lateral sclerosis (ALS). Some of these assemblies can be very dynamic; they condensate and dissolve very quickly. For example, the poly(A)-binding protein (Pab1) of *Saccharomyces cerevisiae*, has been demonstrated to phase separate in vivo and in vitro to form hydrogels in response to physiological heat stress [12]. In this context, phase separation of Pab1 is suspected to function in regulating translation of heat stress-related mRNAs [13]. Like Pab1, the yeast Sup35, a translation termination factor, is also capable of forming liquid/gel-like condensates in vivo during energy depletion-induced pH stress which completely dissolves once conditions return to normal [13]. The dynamic nature of these structures is very interesting for rapid response to quickly changing environments. Interestingly, some of these structures can in addition lock into a stable prion state which is an appealing property for a protein complex to establish and maintain cellular memory. Prions are self-templating, altered, heritable forms of normal cellular proteins which are often associated with neurodegenerative diseases and more recently adaptation to environmental changes [14,15]. Phase separating proteins and prions are both enriched in low complexity regions biased for specific amino acids. The difference, however, is that prions form stable structures while structures arising from phase separation are more dynamic [15].

In this review, we discuss some of the principles of protein phase separation and how cells use phase separation to adapt to stress conditions. We also examine how phase separation can support cellular memory in different organisms.

## 2. Phase Separation Drives the Formation of Membraneless Organelles

Phase separation may involve a single protein (simple coacervation) or it could require two or more proteins or RNA forming a complex (complex coacervation) [16]. A feature often found in the sequence of proteins that undergo phase separation is the presence of intrinsically disordered regions (IDRs) [17]. They are characterised by low complexity regions enriched in repetitive amino acid sequences or motifs [18]. These motifs are multivalent and allow the formation of a network of crosslinking caused by specific motifs within the same protein (Figure 1) [19]. Multivalency refers to specific intermolecular interactions resulting from the presence of multiple domains or motifs within a protein [16]. Some studies have described proteins with multivalency as quinary structures, which are protein structures that are shaped and selected by evolution rather than randomly formed, misfolded aggregates [20,21,22]. Because such repeats in IDRs are enriched with amino acid side chains possessing biased properties (uncharged, charged or aromatic), specific secondary structure driving forces such as hydrogen bonding, dipole–dipole, pi–pi interactions and pi–cation interactions lead to a variety of contacts within the same protein structure [23,24].

IDRs enriched in positively charged amino acids (arginine/glutamine) have the tendency to bind negatively charged RNA and phase separate in the process [16]. For example, Fused in sarcoma (FUS), an RNA-binding protein involved in a range of essential processes including transcription, splicing, mRNA transport and translation, undergoes reversible phase separation in vitro between a dispersed, liquid droplet or hydrogel state under low salt concentration. Mutations in FUS induce pathological protein assemblies which are associated with diseases such as familial amyotrophic lateral sclerosis (fALS) and frontotemporal lobar degeneration (FTLD). This shows why it is important to investigate the structure and dynamics of IDR-containing proteins [24,25]. FUS contains an RGG domain enriched in arginine/glycine which displays a high content of beta-strand structure [26]. The presence of RNA in solution with the FUS protein accelerates its self-assembly, because the RNA-binding domain provides a scaffold for self-assembly of the protein [27]. Together, IDRs, RGG domains of FUS and RNA form essential components of stress granules and concomitantly, multivalency influences the capability of a protein to phase separate [27].

Many proteins and in many organisms across kingdoms contain IDRs. For example, the budding yeast *S. cerevisiae* proteome contains over 20% of proteins with IDRs, and in silico screening of IDR proteins in 31 eukaryotic genomes showed that 52–67% have long (at least 40 consecutive residues) IDRs [28,29,30,31,32]. However, some proteins are more prone to phase separate than others. Currently, it is unclear what really determines the phase separation tendency of an IDR-containing protein solely from their sequences. An answer may come from analysis of the steady-state phase separation of Whi3 in a multinucleate organism such as *Ashbya gossypii*. Whi3 is an mRNA binding protein involved in regulating nuclear division and polarity by interacting with various mRNAs [33]. These interactions affect the structural organisation of Whi3, which consequently allows it to regulate these processes [33]. Whi3 forms droplets in vitro and in vivo. In vivo, phase separation of Whi3 is largely promoted by two mRNAs *CLN3* and *BNI1* and interestingly, the biophysical properties of the droplets are specific to the bound mRNA [33]. Local concentrations of Whi3, *CLN3* mRNA and *BNI1* mRNA could therefore serve as a way to regulate various droplets associated with different cellular processes at distinct locations within the same cytoplasm. This could be a mechanism by which constitutive phase separation and a switch between two states is regulated. Phosphoregulation may be another reason that could be responsible for specifying the function of various Whi3 condensates [34]. Thus, it is important to note that the assembly of membraneless organelles is the collective effect of multivalency and other factors such as RNA which can aid or expedite their assembly. This may underline that multivalency is an essential criterion for phase transition and that IDRs are only the tip of the iceberg of protein sequences promoting phase separation [35,36].

As a result of different combinations of intermolecular interactions, multivalency can give rise to a range of phase separated structures. Here, two case studies in which disparate structures arise from two different IDR-containing proteins can be considered: the regulation of P granules assembly by the RNA helicase LAF-1 in *C. elegans* and a protein involved in ALS, TDP-43. P granules are *C. elegans* germline granules consisting of RNA and protein condensates within the cell and are key in determining the properties of germ cells [37,38]. Phase separation of P granules is induced by the *N*-terminal low-complexity RGG domain of LAF-1 [3]. The fact that the intermolecular interactions governed by the LAF-1 RGG domain are influenced by salt concentration underscores that such interactions are weak enough to be destabilised by an increase in electric charge [3]. This is important as these weak interactions in liquid droplets formed by LAF-1 render them reversible, unlike TDP-43 aggregates.

TDP-43 is an aggregation-prone protein which largely influences the onset of sporadic ALS and frontotemporal dementia, owing to the presence of ALS-linked mutations in its C-terminus [39]. Similar to LAF-1, TDP-43 contains an arginine/glycine rich domain. However, unlike LAF-1, TDP-43 has the propensity to form solid deposits. The presence of stress granules has been shown to enhance TDP-43 aggregate formation due to the sequestration of proteostasis factors such as HDAC6. Indeed, HDAC6 inhibition correlated with an increase in TDP-43 size [40]. The enhancement of TDP-43 aggregation by stress granule association has also previously been proposed to be a result of increased ubiquitylation and reduced splicing activity [41]. Remarkably, the point mutations found in ALS patients accelerate the formation of TDP-43 solid inclusions in vitro [39]. Interestingly, different morphologies were also observed depending on the amino acid mutated, such as defined filamentous or amorphous structures [39]. The fact that IDRs can induce different types of assemblies could be explained by the conditions; it is possible that in physiological conditions, these proteins favour one form over another depending on, for example, the presence of RNA or a protein concentration threshold. Thus, each type of structure may have specific assembly requirements. It is not surprising that similar IDRs lead to a range of structures, as this is characteristic of multivalency and weak interactions in quinary structures, as well as the pleiotropy in some protein assemblies [19,22,42]. However, the biochemistry of such structures is likely to be far more complicated inside cells than what is currently experimentally observed, and there may be many more factors influencing them.

## 3. Phase Separation as an Adaptation Mechanism in Cells

Specific environmental cues including temperature, pH, osmotic and nutrient changes trigger phase separation of proteins and RNA. Here, we discuss a series of adaptation mechanisms that rely on protein phase separation.

### 3.1. Polyadenylate Binding Protein (Pab1) Droplet Formation in Temperature Sensing

The temperature threshold at which most cellular processes are able to be carried out is limited. Beyond these thresholds, physiology begins to deteriorate as a consequence of protein denaturation and disruption of cell membrane integrity. Therefore, living organisms need to execute an ensemble of adaptation mechanisms to try to survive to temperature stress. Stress granules (SGs) are formed in response to unfavourable changes in environmental conditions including temperature. While numerous stress-granule-associated proteins have been identified, Pab1 is unique in its exceptional temperature-sensitivity. Indeed, Pab1 displays a thermal-responsive self-assembly rate 160-fold higher than typical biological reactions [12]. Like many SG-associated proteins, Pab1 has been hypothesized to upregulate mRNA translation by forming higher-order protein assemblies (Figure 2) [12]. The structural requirements for Pab1 to phase separate as an adaptive response to heat shock has been previously probed in vitro, where the significance of multivalency in the context of Pab1 demixing has been highlighted. It was found that while the P domain of Pab1 (the proline-rich IDR) can modulate the extent of phase separation, it is not necessary for phase separation to occur. The deletion of the proline-rich domain of Pab1 showed reduced phase separation [43]. However, the additional knockouts of 3 of its RRMs (RNA-recognition motifs) exhibited the greatest increase in the temperature boundary required for its phase separation [43]. This implicates that in the absence of each of its six domains, Pab1 was still able to phase separate at different temperatures, but absence of the RRMs showed the most impairment to de-mix. Therefore, the IDR of Pab1 is not required for phase separation and the RRMs contain major molecular determinants for demixing. This suggests that Pab1 demixing is more reliant on multivalent interactions of the RRM domain with the IDR serving as an additional tuner, rather than the IDR being the key player.

While being an unparalleled temperature-sensing system, the P-domain of Pab1 is highly evolutionarily conserved across fungal species [1,12]. Remarkably, the frequency of usage of aliphatic residues within the domain is reflected by a fitness advantage. Amongst numerous fungal species, proline-rich regions are almost identical, while alternating patterns between aliphatic residues such as methionine, valine and isoleucine is observed [12]. Interestingly, a negative relationship was found between the frequency of aliphatic amino acids and their hydrophobicity in the interchangeable sections of the P-domain across these species [12], while this trend was different albeit consistent across the rest of the protein sequence, the yeast proteome and disordered regions. The composition of these flexible regions of the P-domain has been naturally selected according to hydrophobicity; Pab1 phase separation is adaptive and has been fine-tuned on an evolutionary time scale. From this, we can conclude that Pab1 acts as a temperature sensor by using its RRMs, and to a lesser extent its P-domain, to tailor mRNA translation levels according to temperature changes within and around the cell through phase separation.

### 3.2. pH Sensing with Poly-Uridylate Binding Protein (Pub1) and the Cytoplasm

Temperature change in the cell is often coupled with fluctuation in pH [43]. Stress response pathways overlap in both conditions, such as the formation of stress granules. Along with Pab1, the protein Pub1 is also a canonical component of stress granules [45] and as the name suggests, it is also an RNA-binding protein [46]. In situations of glucose starvation, cellular ATP levels drop and the proton-exporting activity by ATPases is reduced, which in turn induces a decrease in cytosolic pH [47]. In these conditions, Pub1 forms condensates and acts as a pH sensor [48].

Is there any common ground in proteins that sense stress? Both Pab1 and Pub1 are SG-related and RNA-binding proteins and the efficiency of phase separation in both proteins is perturbed by the presence of RNA [12,48]. Moreover, in both cases, the RRM domains play a significant role in influencing phase separation. In another study, imaging of cells expressing full-length Pub1, Pub1-RRM (Pub1 with only RRMs) or Pub1-LC (Pub1 with only the low-complexity domain) showed that only the former two variants formed condensates after energy depletion-induced pH change [48]. Considering the data on the P domain in Pab1, such IDRs/LC domains seem to serve a secondary role in regulating phase separation via the RRMs, which act as the primary modulators [48]. While further studies are needed to pinpoint the canonical phase separating features of SG-proteins, these studies on Pab1 and Pub1 tell us that their IDRs act to regulate solubility of the protein which provides specificity of phase separation morphology. Such morphologies are likely to be evolutionarily refined to detect different thresholds of stress.

In more extreme cases of pH stress, a widespread phase transition can also occur in the cell as a result of entry into dormancy [49]. This shifts the cell into a state of cell cycle arrest and reduced metabolic activity involving the packing of proteins into higher order structures, caused by a transition of the cytoplasm from a liquid to a glass-like state upon manipulation of cytosolic pH in yeast and bacteria [49,50,51] (Figure 2). This is a more stringent mechanism of phase separation as it is triggered in response to more extreme environmental stresses. This cytoplasmic freezing has been observed in budding yeast upon reduction of cytosolic pH, which was achieved by depleting 95% of cellular ATP, inducing dormancy. Unlike conventional techniques used such as fluorescence microscopy to discern condensate formation, this study placed its focus on the mechanical stability and mobility of macromolecules in the cell using micro-rheological techniques [49]. To measure the fluidity of the cytoplasm, a bead-like foreign tagged particle was introduced into the cells and its movement within the cell was tracked. It was found that lowering cytosolic pH alone in the presence of glucose (no starvation) is sufficient to display a reduced particle mobility phenotype, thus reduced cytoplasmic fluidity [51]. This reinforces the idea that proteins and macromolecules respond to simple physicochemical cues both on an individual and global scale. Interestingly, upon screening of the isoelectric points of the entire yeast proteome, the majority were found to overlap with the pH value corresponding to conditions of starvation [51]. Because the solubility of proteins decreases abruptly when their surrounding pH converges with their isoelectric point [52,53], it is highly likely that such proteins phase-separate to form higher-order assemblies in instances of energy depletion. As a result, the cell enters a state of dormancy when this occurs in a widespread manner under precipitous conditions, which could explain the overall change from fluid to solid-like state of the cytoplasm [51]. This is biologically relevant in terms of phase separation, as cytoplasmic freezing appears to be a mechanism by which proteins, as well as RNA, are reorganised into such a state that likely also modulates translational activity to reduce metabolism.

Molecular crowding was found to also play a role in the cytoplasmic freezing of yeast cells in conditions of glucose starvation, ultimately leading to a decrease of mechanical fluidity of cellular components [49]. Bacterial cytoplasmic freezing has been recorded as a mechanism to organise cellular components in a size-dependent manner to regulate metabolic activity [50]. This demonstrates that glass-like states may be induced by a variety of stimuli and it begs the question of whether this event is confined to unicellular organisms, or in other words, do multicellular organisms regulate metabolic activity through cytoplasmic freezing? While it has been reported that cells of metazoans such as the marine brine shrimp also enter dormancy in a pH-dependent manner [54], it is unclear whether they do so with the same mechanism. However, it is likely that single cells utilize stimuli-induced changes in global material properties of the cytoplasm more commonly than multicellular organisms to sense and adapt to simple physicochemical changes in an individual manner. This is probably owing to the fact that multicellular organisms have alternative routes of sensing stress, such as the sensing of temperature stress in *C. elegans* through thermosensory neurons to regulate metabolic activity and initiate an organismal response [55].

### 3.3. Osmotic Shock Foci (OSF) Formation in Osmotic Stress

A ubiquitous environmental challenge that cells experience in all organisms is a change in extracellular ion/salt concentration, which often leads to hyperosmotic shock to cells [56]. Consequently, this causes cell shrinkage and water to move out of the cell [57]. Similar to a change in pH, this process also disrupts the electrostatic charge of the intracellular area, which can concomitantly influence the multivalent interactions (such as protonation of side chains) within proteins and affect solubility, inducing phase separation.

To investigate proteins that phase separate in response to osmotic shock in yeast, the formation of P-bodies and sequestering of chaperones has been recently characterised; these proposed liquid-liquid phase separated droplets were termed as OSFs (osmotic shock foci) [57]. Highly dynamic and reversible OSFs that were found to be the chaperones Ssa1, Hsp104 and Hsp42 formed in vivo upon induction of hyperosmotic shock, which disappeared swiftly upon stress removal [56]. The speed and reversibility of this appearance supports the notion that such chaperone OSFs are liquid droplets rather than stable aggregates. Interestingly, along with the reversible P-body proteins Dcp2 and Edc3, the overexpression of other proteins with amyloidogenic domains which have been reported to form stable aggregates (Sup35, Mot3 and Pan1) (Figure 2) [56], were all able to assemble and disassemble swiftly following shock and relaxation respectively. This exhibits a drastically different phenotype to amyloid structures which are reportedly fairly stable given that they usually require solubilizing agents such as guanidine hydrochloride and sodium sulfate to be reversed [58,59]. Moreover, this indicates that the formation of these alternative, unstable liquid-like droplets appears to be favoured over stable amyloid aggregates in specific hyperosmotic conditions.

The main driving factor of phase separation as a result of hyperosmotic stress could be explained by molecular crowding [60]. In another recent study, the protein YAP (yes-associated protein) in mammalian cells was also shown to form liquid droplets seconds after hyperosmotic shock (Figure 2), and to localise to areas of the cell with high concentrations of accessible chromatin domains as a result of cell shrinkage and molecular crowding [60]. YAP is a transcriptional factor that normally binds to enhancers of specific target genes [61]. Phase separation of YAP relies on its IDR (the transcription activation domain), as mutants lacking this domain failed to form droplets under crowding conditions [60]. The fact that liquid droplet formation can be induced directly by crowding implicates that such proteins act as sensors of mechanical stress produced by crowding, thus phase separating and localising specifically to adjust and regulate gene expression. Phase separation could function as a rapid mechanism to respond to mechanical stress, and YAP serves as a starting point to expanding the library of mechanical-sensing phase separating proteins that we already know, such as TAZ and spindroin-like proteins in spider silk [62,63,64].

### 3.4. Sup35 Gelation in Nutrient Deficiency

One of the most well-characterized prion-like proteins in yeast is Sup35, a translation termination factor which, upon inactivation via a conformational switch to its prion form [*PSI*+], leads to STOP codon read-through [13,52]. Recent studies have revealed the reversible, non-prion structure of Sup35, as a result of energy depletion-induced gelation (Figure 2) [13]. Sup35 can also be considered as a stress granule protein due to its ability to sequester with Pab1 after phase separation and to regulate translation termination [52]. It is unclear whether this alternative form of Sup35 directly results in its inactivation and translation read-through. A study on Sup35 has demonstrated that, unlike prion formation, gelation only requires its GTPase catalytic domain which is responsible for translation termination in vivo [13]. Although this is the case, the IDR of Sup35 (NM domain) was shown to be important for governing the droplet properties of Sup35 gels [13]. The fact that the catalytic domain alone was able to form droplets could further suggest that gelation of Sup35 can be achieved solely with the multivalent interactions within this domain rather than requiring the IDR of the NM domain. However, the NM domain is important for rescuing the catalytic domain from potential stress-induced damage by enhancing droplet reversibility. It is important to note that Sup35 prions are also reversible, although this often requires curing with chaperones, while gel structures appear to be reversed simply through a conditional feedback process [13,65].

Sup35 is structurally versatile in that it can phase separate into gel-like structures, but it is also a prion which can form amyloid-like structures. Sup35 is present across many fungal species, but [*PSI*+] is predominantly found in *S. cerevisiae* [66,67]. Interestingly, phase separation of Sup35 is conserved among distant yeast relatives, while this is not the case for prion formation [13]. In contrast to *S. cerevisiae*, Sup35 is unable to propagate [*PSI*+] in *Schizosaccharomyces pombe* in a Hsp104-dependent manner, while both have been shown to be able to form stress-induced Sup35 condensates [13]. However, whether endogenous [*PSI*+] can be propagated in *S. pombe* has not yet been tested and it is possible that prion formation is mediated by other chaperone machineries. This suggests that the temporary phase transition as a stress response mechanism may evolutionarily precede prion propagation and induction. The reason why *S. cerevisiae* possesses numerous prion-forming proteins while only one was found in *S. pombe* so far [68], but forms other condensates, is still unclear. This is especially interesting as *S. cerevisiae* and *S. pombe* both share a wide range of chaperone systems that are responsible for prion propagation [68,69]. The difference in their prion-forming tendencies may be related to asymmetric cell division of *S. cerevisiae*, or the possession of other cytoplasmic determinants in budding yeast that are open to be discovered in the future.

Taking this data into consideration, we can hypothesize that Sup35 phase separation serves a distinctly different role to prion formation due to its induction under specific stress conditions such as nutrient deficiency. Upon recovery to healthy conditions, such condensates return to their soluble state in the cell [13]. Contrarily, Sup35 prion formation has been proposed to occur as a means to maintain a basal level of mRNA translation regulation and to pass on stable aggregates to future generations [52]. This is interesting because Sup35 gelation appears to be a frequently occurring, stress-sensing process that will happen in all cells, while Sup35 prion formation is unlikely to happen. Conversion to [*PSI*+] has been proposed to be a bet-hedging device in terms of translation fidelity, such that prion formation is positively selected for stressful conditions in exchange for a reduced overall fitness [70,71]. Nonetheless, both mechanisms are orchestrated for cell survival in fluctuating environments.

### 3.5. Mitochondrial Antiviral Signalling Protein (MAVS) Fiber Formation in Antiviral Immune Responses

The implementation of phase separating proteins has also been reported in binary decision-making and signal transduction in innate immune response. In humans, the mitochondrial antiviral signalling protein (MAVS) is activated by the viral RNA-detecting protein RIG-1 [44] and polymerizes into amyloid fibers [72] (Figure 2). MAVS contains a CARD (caspase activation recruitment) domain at its *N*-terminus, which belongs to the death domain superfamily. The CARD domain underlies filament formation [73,74,75] and shares features with prion fibers [73]. Formation of filaments is required for the activation of the downstream effectors NF-kB and IRF3 in the RIG-1 immune response pathway [75,76]. Therefore, MAVS amyloid fibers and prion-like behaviour is a physiological mechanism.

Interestingly, the CARD domain found in MAVS is able to functionally replace the NM domain of Sup35 in yeast [44] and remarkably, the NM domain can also function in place of the MAVS CARD domain in an NM-MAVS construct in mammalian cells. NM-MAVS filament formation was promoted by transfecting cells with NM fibers inducing downstream signalling, which was measured by activation of interferon beta 1a [44]. This highlights the remarkable interchangeability of Sup35 NM and MAVS CARD domains, which seem to mostly require filament assembly and self-templating activity. This, contrary to the different faces of Sup35 we have discussed before, may suggest that prion behaviour can be selected for during evolution and is not simply a by-product of phase separation. The relationship between multivalent interactions providing a biophysical basis for phase separation and self-templating activities displayed by prions is a fascinating topic to be addressed in the future.

### 3.6. Std1 and Carbon Source Sensing

*S. cerevisiae* cells are craving glucose as a primary carbon source for the production of energy. The presence of glucose in the culture medium represses the expression of genes involved in alternative carbon source utilisation, a phenomenon termed glucose repression [77]. Cells must therefore fine tune their ability to sense and respond to low glucose levels if they are to survive. Snf1, a heterotrimeric protein kinase of the Snf1/AMP-activated protein kinase family, primarily ensures survival under limiting glucose conditions by activating genes involved in utilisation of sucrose, galactose or ethanol. Snf1 is also involved in cellular developmental processes such as meiosis, sporulation and ageing as well as response to other stress conditions such as oxidative stress, heat stress and regulation of various metabolic enzymes. Activation of Snf1, under glucose limitation is mediated by Std1 [78,79,80,81]. Under glucose replete condition, Std1 undergoes phase separation and localises to cytoplasmic foci. When glucose is limiting, foci formed by Std1 dissolve, releasing Std1 which can translocate to the nucleus and activate Snf1. Through this mechanism, cells are able to respond appropriately to limiting glucose levels and probably buffer noise in the concentration or readout of the concentration of sugar [80]. Phase separation is therefore a very interesting mechanism for both sensing and adapting to stresses, as demonstrated by the few examples we have presented here. Interestingly, Std1 when over-expressed in *S. cerevisiae* promotes the formation of [*GAR+*] prion, generating an adaptive phenotype where the cells are not reliant on glucose as primary energy source [82]. Again, just as in the case of Sup35, this demonstrates how the same protein can promote the formation of structurally and physically distinct structures depending on different environmental cues.

## 4. Phase Separation and Cellular Memory

In addition to adaptation, there have been examples of protein-based memory which may rely on phase separation. Whi3, in *S. cerevisiae*, is involved in encoding memory of deceptive courtship and cytoplasmic polyadenylation element-binding proteins (CPEB) are involved in long term potentiation during courtship in *Drosophila*.

There are two mating types in haploid yeast cells, *MAT*a and *MAT*α, which release pheromone as a and α-factor respectively. Mating factors are sensed via the cell surface receptors Ste2 (*MAT*a) and Ste3 (*MAT*α) [83,84,85]. Cells exposed to α-factor respond by arresting in the G1 phase of the cell cycle and developing cytoplasmic projections (called a shmoo) in order to grow towards the source of pheromone, which is supposedly a mating partner. However, when there is no mating partner, cells escape pheromone arrest and proceed to division [86]. This type of failed mating attempts can arise when cells cannot reach their mating partner in time, which could happen for example if several cells try to fuse with the same one [83]. After escape from pheromone arrest, cells maintain a memory of these deceptive mating encounters in the form of Whi3 condensates into super-assemblies. Therefore, like Sup35, Whi3 appears to be able to form or join very different condensates because it was found in stress granules, super-assemblies and age-induced aggregates [87,88]. Whi3 is an mRNA binding protein which regulates cell size and cell cycle progression during the G1 phase of the cell cycle [89,90]. These super-assemblies asymmetrically partition into mother cells when the cells divide after escaping pheromone arrest (Figure 3). Indeed, mother cells do not shmoo any longer in the presence of pheromone, but daughter cells emanating from these refractory mothers do not inherit this memory and shmoo as soon as they are born [83,87,91].

Whi3 from *S. cerevisiae* possesses low complexity regions rich in Q/N residues, which have been shown to mediate super-assemblies formation and memory [83,90]. Since Whi3 super-assemblies are asymmetrically inherited, we suspect that such asymmetric segregation could be a powerful mechanism to establish cell fate and maintain asymmetries. One observation however is that these self-templating proteins have the tendency to adopt detrimental conformations during ageing. For example, some variants of Sup35 accumulate specifically at age-induced protein deposits [87]. Conformational flexibility offered by the PrD comes at a cost when cellular physiology changes during ageing. For example, pH is not regulated or controlled as tightly as in young cells and this may have as a consequence that conformationally flexible proteins are suddenly locked into a more stable and solid state. Whi3 also forms foci during ageing that result in a pheromone insensitivity of old yeast cells [92]. Interestingly, cells harbouring a mutant form of Whi3 (*whi3-ΔpQ*) which do not adopt the super-assembly conformation have been shown to live slightly longer than wild type cells [87,93]. It is not clear whether the asymmetric segregation of Whi3 super-assemblies in mother cells is a mechanism to induce ageing and limit lifespan. Furthermore, it is not clear whether the variants of Sup35 which accumulate at age-induced deposits have a function or not. These questions are still open for investigation.

The involvement of prion-like proteins in formation and maintenance of memory is not a specificity of budding yeast. Cytoplasmic polyadenylation element-binding proteins (CPEBs) are mRNA binding proteins that exists in different cells types in a range of organisms such as sea slugs (*Aplysia californica*, AsCPEB), flies (*D. melanogaster*, Orb2) and mice (CPEB3) where they regulate protein synthesis. Isoforms of CPEBs found in neurons particularly differ from those found in other cell types by the fact that their *N*-terminal is rich in glutamine which is a typical characteristic of yeast prion domains [94,95]. CPEBs exist in two different conformational states, a monomeric form and a prion-like self-sustaining amyloidogenic aggregated form that behaves like a prion [95]. Aggregated states of CPEBs have been linked to persistence and regulation of memory [96,97]. For example, in *Drosophila*, an already mated female fly refuses to mate again. After prolonged rejection, a male fly learns to suppress its desire to mate with already mated females, but not with virgin ones, for days because it develops a long-term memory that it had been rejected before [98]. Flies which have significantly reduced aggregation of Orb2 also have impaired retention of long-term memory of previous rejection. Orb2 has two isoforms, a shorter, very rare and poorly expressed Orb2A and a longer, abundant and constitutively expressed Orb2B. Orb2A is required for aggregation of Orb2B, a process necessary for the formation of long-term memory. Both isoforms possess prion-like sequences in their *N*-terminal domain [98]. In vitro studies have shown that the aggregated form of Orb2 is an activator of translation while the monomeric form acts as a repressor of target genes associated with long term memory [99]. The monomer binds to the CG13928 protein, which is responsible for repression, while the oligomer binds to CG4612, which contributes to activation (Figure 4) [99]. This observation is not only limited to Orb2. Interestingly, this phenomenon has also been reported for AsCPEB where the active state is the aggregated prion state when it actively binds mRNA in vitro [95]. In neurons, AsCPEB exclusively form prion-like punctate structures where they also regulate long term synaptic facilitation. It was suggested that these structures may have formed from protein–protein interactions [97]. Protein–protein interactions between one or more proteins also promotes liquid-liquid phase separation [16]. To investigate the exact role of the prion form of AsCPEB in vivo, Kausik et al. [96] proceeded to overexpress AsCPEB in neurites and observed formation of puncta outside synaptic areas which had no effect on normal synaptic functions but retained ability to bind mRNAs. They therefore suggested that these prion forms may actually play other active roles in this conformation. Conformational switches in classical prions normally induce loss of function phenotype [73]. It appears therefore that apart from being significant in phase separation, conformational switches may actually uncover activity in certain prion-like proteins.

This observation opens up a new paradigm of looking at how Whi3 regulates memory in yeast. Our current understanding is that conformational change to the super-assembly form transforms Whi3 from an active inhibitor of *CLN3* mRNA translation to an inactive protein. However, it could well be that super-assembled Whi3 is activating or enhancing translation of *CLN3* mRNA. Altogether, this could suggest that conformational switches of prion-like proteins may have a conserved functional role.

Another parallel that can be drawn between Whi3 and AsCPEB is that both proteins undergo conformational changes specifically in the presence of pheromone and serotonin respectively. This suggests that different memory encoding substances may be activated differently in a manner that is not characteristic of prion proteins. We propose that, added to phase separation, prion-like characteristics such as self-templating conformations of these proteins may be necessary for memory. Confinement of this activity to the mother cells or activated synapse suggests that there is some form of control. The mechanism behind this, however, is yet to be uncovered.

## 5. Conclusions

What precise role could phase separation play in ageing? Is there possibly an interplay between ageing and memory acquisition? Ageing is a gradual decrease in the physiology of cells with time or number of divisions. During ageing, cells have difficulties to deal with accumulated aberrant protein aggregates which results partly from weakened mitochondrial activity. Ageing also affects proper regulation of gene expression which in turn can result in the distortion of protein concentrations and stoichiometry. Therefore cells have to cope with the increasing need to repair or degrade damaged proteins that accumulate as they undergo complex metabolic and physiological activities [100,101]. The overall consequence of this is a deviation from physiological states and ultimately death. Changes in RNA concentration has been shown to affect phase separation of RNA-binding proteins and subsequently the properties of biomolecular condensates [33]. While IDR-bearing proteins such as FUS and TDP-43 have been shown to fuse into normal stress granules, mutants of these proteins contribute to irreversible phase transitions of stress granules into pathological stress granules which are hallmarks of age-related diseases such as Alzheimer’s disease [101,102]. The clearance of older stress granules as well as pathological stress granules involves the autophagic machinery [103,104]. Defective autophagic machinery has been linked with increase in prion formation and neurodegenerative diseases such as ALS [105,106,107]. It is however not clear how pathological stress granules evade cellular clearance machineries. We have noted previously that Whi3 and Sup35 aggregate into a range of different structures including age-induced deposits. It seems that IDRs have evolved to allow for structural flexibility which enable proteins to phase separate into different assemblies where their exact function may not yet be very clear.

In summary, we have highlighted how certain prion-like proteins, driven by multivalency, undergo phase separation and adopt different physical states under various stress conditions, including those implicated in memory establishment. We suspect that phase separation could be a mechanism by which single cell organisms such as yeast and bacteria, as well as a number of multicellular higher organisms, cope with stress conditions on a cellular scale. This process has also been refined by natural selection across these organisms on the primary sequence level for each IDR-bearing protein in specific stress scenarios, giving a spectrum of structures and components involved. While we have described several well-studied stress-response systems, we predict that findings on phase separation in memory acquisition and possibly ageing such as for Whi3 and CPEBs are only at the beginning of their emergence.

## Figures and Tables

**Figure 1 cells-09-01302-f001:**
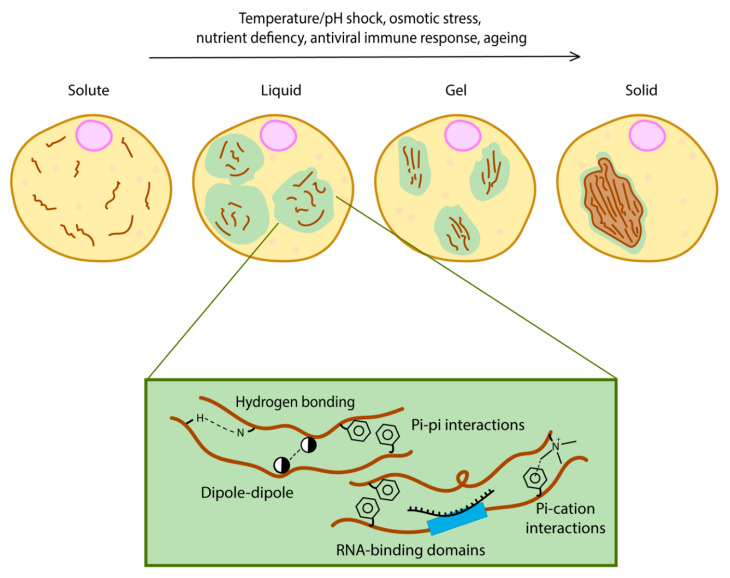
Different levels of phase separation. Multivalent interactions between intrinsically disordered regions (such as hydrogen bonding, pi–pi interactions, pi–cation interactions and dipole–dipole interactions) are necessary for phase separation.

**Figure 2 cells-09-01302-f002:**
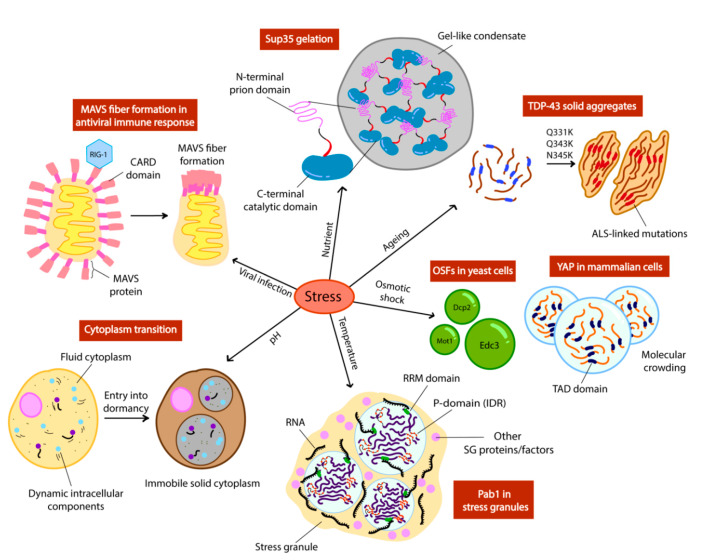
Different stress signals trigger phase separation of intracellular components in yeast (OSFs, Pab1, cytoplasm, Sup35) and mammalian cells (MAVS, TDP-43, YAP). Schemes adapted from [12,14,44].

**Figure 3 cells-09-01302-f003:**
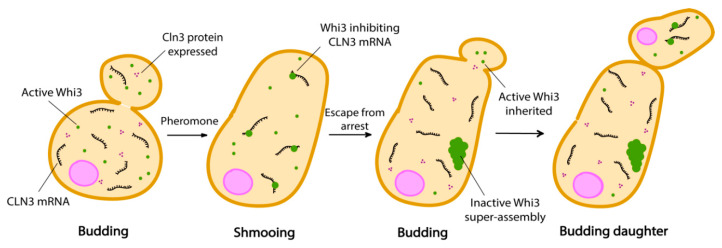
The pheromone refractory state in budding yeast is propagated by Whi3 super-assembly formation. Cln3 protein expression is constitutive when *CLN3* mRNA is free from inhibition. Inactive Whi3 is confined in the mother cell, while daughter cells containing active Whi3 continue to shmoo.

**Figure 4 cells-09-01302-f004:**
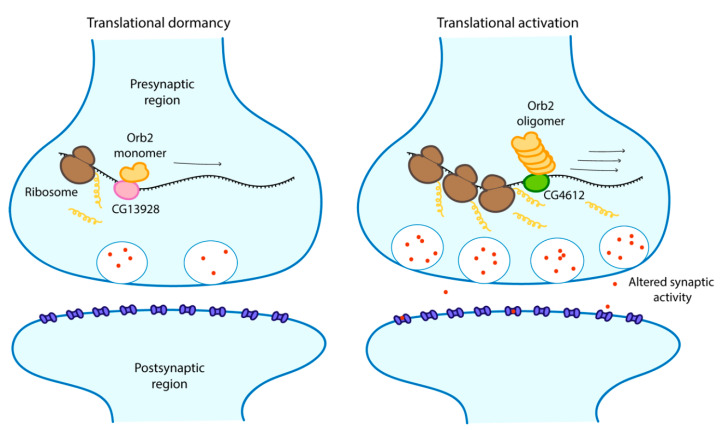
The repression and activation of mRNA translation by Orb2 monomer and oligomer respectively, resulting in altered synaptic activity and memory stabilisation. Scheme adapted from [99].

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
