# Peer review of "Protein Phase Separation during Stress Adaptation and Cellular Memory"

_cells, 2020, doi:10.3390/cells9051302_

Round 1
Reviewer 1 Report
This nice review by Lau and colleagues summarizes the emerging field of protein phase separation and its potential role undergirding cellular adaptation, and even ‘memory.’ Overall it is well written and suitable for the readership of the journal. My comments below are minor suggestions to enhance interest and readability.
Major:
- The introduction to this review hints at the relationship between phase separation and prion behavior, and I think it would be nice to extend that thread a bit more fully throughout the manuscript. This comes already a bit in the context of Sup35, but it would be good to also discuss this for [GAR+] in the context of Std1 phase separation. The same proteins are involved, but the Std1 condensates clearly are not [GAR+], just as the Sup35 gels are not [PSI+]. I recommend being nuanced in these discussions; in some places in the manuscript it is presently suggested that prions are irreversible (e.g. line 300). Although this is comparatively true relative to biomolecular condensates without any feedback process, there is now ample evidence prions can be eliminated, and induced, by a variety of mechanisms.
- In Figure 1 more distinction between gel and solid depictions would be helpful to the reader. Also, the fonts in the lower green box are too small to read in the current version.
- The authors have made a good inventory of forces driving phase separation. It be a good idea to include cation-pi interactions too: e.g. https://pubmed.ncbi.nlm.nih.gov/29677515/
- The authors suggest (as has been widely suggested by some in the literature) that acidification could drive of gelation and vitrification (e.g. line 168) in response to starvation, etc. Is pH 5.7 achieved under these circumstances? This seems extreme, and it might be good to expand here.
- For the Sup35 discussion from lines 264 to 295, I would emphasize that the effect on translation has not been fully investigated for the gels. Also, there is an important distinction between the statements made on line 287 about the inability of S. pombe Sup35 to act as a prion vs. what was actually shown in the literature, which has oft been repeated. S. pombe Sup35 cannot propagate [PSI+] in way that depends on S. cerevisiae Hsp104 dependent when both proteins are expressed in cerevisiae. Whether it can act as a prion in the endogenous organism, or one that depends on another chaperone, has not been tested.
Minor comments:
- Italicize species names throughout
- Line 38: add “yeast” Sup35 for context.
- Line 41. Mention that the stable state is a prion.
- Line 59: Glutamine is usually charged unless the pKa is perturbed in some way.
- Line 86 “effects” should be “effect”
- Line 438-439: This is a very strong statement about the origins of aging. More circumspect wording would be appropriate
- I suggest double checking the references. Something seems amiss. For example, number 18 is the title of an Alan Drummond paper and has the right title, journal and page numbers, but the wrong author list. Several others also seemed like they could be off.
Author Response
Reviewer 1
This nice review by Lau and colleagues summarizes the emerging field of protein phase separation and its potential role undergirding cellular adaptation, and even ‘memory.’ Overall it is well written and suitable for the readership of the journal. My comments below are minor suggestions to enhance interest and readability.
We thank reviewer 1 for her/his comments which have improved the manuscript.
Major comments:
- The introduction to this review hints at the relationship between phase separation and prion behavior, and I think it would be nice to extend that thread a bit more fully throughout the manuscript. This comes already a bit in the context of Sup35, but it would be good to also discuss this for [GAR+] in the context of Std1 phase separation. The same proteins are involved, but the Std1 condensates clearly are not [GAR+], just as the Sup35 gels are not [PSI+]. I recommend being nuanced in these discussions; in some places in the manuscript it is presently suggested that prions are irreversible (e.g. line 300). Although this is comparatively true relative to biomolecular condensates without any feedback process, there is now ample evidence prions can be eliminated, and induced, by a variety of mechanisms.
We have now added text to mention that Std1 condensates are different from [GAR+]. We have also nuanced our discussion on the irreversibility of prions in the review and removed the term “irreversible” line 300 to keep only “stable”.
- In Figure 1 more distinction between gel and solid depictions would be helpful to the reader. Also, the fonts in the lower green box are too small to read in the current version.
Figure 1 was modified to depict better the difference between gel and solid. Font in all the figures was increased.
- The authors have made a good inventory of forces driving phase separation. It be a good idea to include cation-pi interactions too: e.g. https://pubmed.ncbi.nlm.nih.gov/29677515/
We would like to thank the reviewer for pointing this out. We have added the interaction and the reference.
- The authors suggest (as has been widely suggested by some in the literature) that acidification could drive of gelation and vitrification (e.g. line 168) in response to starvation, etc. Is pH 5.7 achieved under these circumstances? This seems extreme, and it might be good to expand here.
We have removed the mention of pH 5.7 to reduce experimental details and did not discuss further whether this value is reached during stress.
- For the Sup35 discussion from lines 264 to 295, I would emphasize that the effect on translation has not been fully investigated for the gels. Also, there is an important distinction between the statements made on line 287 about the inability of S. pombe Sup35 to act as a prion vs. what was actually shown in the literature, which has oft been repeated. S. pombe Sup35 cannot propagate [PSI+] in way that depends on S. cerevisiae Hsp104 dependent when both proteins are expressed in cerevisiae. Whether it can act as a prion in the endogenous organism, or one that depends on another chaperone, has not been tested.
We have amended the text according to the suggestion of this reviewer both on the effect of gelation of Sup35 on translation and on Sup35 in S. pombe.
Minor comments:
- Italicize species names throughout
Done
- Line 38: add “yeast” Sup35 for context.
Added
- Line 41. Mention that the stable state is a prion.
Done
- Line 59: Glutamine is usually charged unless the pKa is perturbed in some way.
Glutamine is mentioned as charged.
- Line 86 “effects” should be “effect”
corrected
- Line 438-439: This is a very strong statement about the origins of aging. More circumspect wording would be appropriate
Ageing definition was reworded.
- I suggest double checking the references. Something seems amiss. For example, number 18 is the title of an Alan Drummond paper and has the right title, journal and page numbers, but the wrong author list. Several others also seemed like they could be off.
Reference was corrected.
Reviewer 2 Report
In their review, the authors describe regulation and function of phase separation in the cellular stress response and memory formation. While the review addresses interesting and highly relevant phenomena and summarizes a lot of the available knowledge in the field, the review rather reads as an extensive collection of experiments/results, but is less successful in providing easily accessible information and a synthesized interpretation for non-experts. Thus, the review rather serves as a highly detailed summary of facts related to phase separation in the context of the cellular stress response and memory formation, but fails to provide the necessary context. Many of the critical terms required for understanding the biology behind the described experiments are never defined. For example, although the review aims to highlight physical properties of prions and how this can contribute to cellular memory, the term prion is never defined, nor is the difference in the properties of a prion relative to other phase separating proteins.
Same for FUS, discussing this protein is highly relevant for understanding potential contributions of phase separation during disease, but FUS is never introduced other than by its intrinsic biophysical properties.
Along the same lines, the authors start their introduction by describing the different phase separated membrane less organelles and describe their highly dynamic nature, but use the in vitro properties of an otherwise non introduced protein, Sup35, as a demonstration for the dynamics of the organelles. Instead, the dynamics of the membrane less organelles that are discussed in this section should be highlighted with an in vivo example, and, once relevant, Sup35 should be introduced mentioning the species it originates from and a cellular function/context in which phase separation of this protein is relevant.
Another example where providing too much detail impairs readability of the review for non-expert readers, line 137: “This is highlighted by the fact that all single-domain deletions still phase separate by 50°C despite having different effects on the temperature to pH ratio required for de-mixing,» While the description reflects the actual experiment, in a review I would rather hope to get an interpretation of the data that can be understood without actually knowing the primary literature and specific experiment.
Some phrases would require rewriting to be clear/accurate.
Example: Line 196 “where the area displaced by the tagged particle in lower pH growth media (around 5.5) was considerably less than that in pH 7.7 « is not an appropriate description of a mean squared displacement analysis.
Line 334: “Snf1, a transcriptional activator of genes under glucose repression.” The description of the Snf1/AMPK kinase as a transcriptional activator does not quite reflect its cellular function.
In summary, I believe that the review would greatly profit from reducing unnecessary experimental detail and focussing on introduction of the terms and key proteins to provide context as well as the interpretation of the data rather than actual results. Focusing on fewer critical proteins (or for example fewer stresses) than extensively listing a large number should be considered.
Author Response
Reviewer 2
In their review, the authors describe regulation and function of phase separation in the cellular stress response and memory formation. While the review addresses interesting and highly relevant phenomena and summarizes a lot of the available knowledge in the field, the review rather reads as an extensive collection of experiments/results, but is less successful in providing easily accessible information and a synthesized interpretation for non-experts. Thus, the review rather serves as a highly detailed summary of facts related to phase separation in the context of the cellular stress response and memory formation, but fails to provide the necessary context. Many of the critical terms required for understanding the biology behind the described experiments are never defined. For example, although the review aims to highlight physical properties of prions and how this can contribute to cellular memory, the term prion is never defined, nor is the difference in the properties of a prion relative to other phase separating proteins.
We thank reviewer 2 for her/his comments. We have added definitions of many terms in the text and we have removed many experimental details.
Same for FUS, discussing this protein is highly relevant for understanding potential contributions of phase separation during disease, but FUS is never introduced other than by its intrinsic biophysical properties.
We have added an introduction on the function of FUS.
Along the same lines, the authors start their introduction by describing the different phase separated membrane less organelles and describe their highly dynamic nature, but use the in vitro properties of an otherwise non introduced protein, Sup35, as a demonstration for the dynamics of the organelles. Instead, the dynamics of the membrane less organelles that are discussed in this section should be highlighted with an in vivo example, and, once relevant, Sup35 should be introduced mentioning the species it originates from and a cellular function/context in which phase separation of this protein is relevant.
Function of Sup35 has now been introduced.
Another example where providing too much detail impairs readability of the review for non-expert readers, line 137: “This is highlighted by the fact that all single-domain deletions still phase separate by 50°C despite having different effects on the temperature to pH ratio required for de-mixing,» While the description reflects the actual experiment, in a review I would rather hope to get an interpretation of the data that can be understood without actually knowing the primary literature and specific experiment.
This has been changed.
Some phrases would require rewriting to be clear/accurate.
Example: Line 196 “where the area displaced by the tagged particle in lower pH growth media (around 5.5) was considerably less than that in pH 7.7 « is not an appropriate description of a mean squared displacement analysis.
We have changed wording here.
Line 334: “Snf1, a transcriptional activator of genes under glucose repression.” The description of the Snf1/AMPK kinase as a transcriptional activator does not quite reflect its cellular function.
This has been changed.
In summary, I believe that the review would greatly profit from reducing unnecessary experimental detail and focussing on introduction of the terms and key proteins to provide context as well as the interpretation of the data rather than actual results. Focusing on fewer critical proteins (or for example fewer stresses) than extensively listing a large number should be considered.
Please find below the list of changes we have made to adjust the manuscript according to the suggestions of both reviewers.
Line 19: “s” deleted from focussing
Line 29: C. elegans italicised
Line 38-41: Added “For example, the poly(A)-binding protein (Pab1) of Saccharomyces cerevisiae (S. cerevisiae), has been demonstrated to phase separate in vivo and in vitro to form hydrogels in response to physiological heat stress [12]. In this context, phase separation of Pab1 is suspected to function in regulating translation of heat stress-related mRNAs [1,13]” to introduce a protein which undergoes phase separation in vivo first.
Line 41-42: Added “Like Pab1” “yeast” and “a translation termination factor”, to put the statement in context
Line 45: Added “prion” to describe the stable state
Line 47-51: Added “Prions are self-templating, altered, heritable forms of normal cellular proteins which are often associated with neurodegenerative diseases and more recently adaptation to environmental changes [15]. Phase separating proteins and prions are both enriched in low complexity regions biased for specific amino acids. The difference, however, is that prions form stable structures while structures arising from phase separation are quite dynamic [16]” to define a newly introduced term.
Line 61-63: Added “Multivalency refers to specific intermolecular interactions resulting from the presence of multiple domains or motifs within a protein [13].”
Line 64: Replaced “refer to” with “are”
Line 67: Added “and pi-cation interactions”
Line 70-76: Added “Fused in sarcoma (FUS), an RNA-binding protein involved in a range of essential processes including transcription, splicing, mRNA transport and translation, undergoes reversible phase separation in vitro between a dispersed, liquid droplet or hydrogel state under low salt concentration. Mutations in FUS induce pathological protein assemblies associated with diseases such as familial amyotrophic lateral sclerosis (fALS) and frontotemporal lobar degeneration (FTLD). This shows why it is important to investigate the structure and dynamics of IDR-containing proteins [24,25]” to introduce FUS and its significance to pathological conditions.
Line 77: Replaced “which was found to have” with “displays”
Line 88: % deleted
Line 95: Deleted “has been shown to”
Line 97: “were” replace by “are”
Line 100-102: Phosphoregulation may be another reason that could be responsible for specifying the function of various Whi3 condensates
Line 102: Additional ref added [34]
Line 102: “s” deleted from effects
Line 106-107: Sentence rephrased to “As a result of different combinations of intermolecular interactions, multivalency can give rise to a range of phase separated structures” for clarity.
Line 115: Added “formed by”
Line 118: deleted “for instance Q331K, Q343R and N345K”
Line 119: Deleted “ly” from similarly
Figure 1: Increased text size and modified depiction of solid aggregates
Figures 2-4: Increased text size
Line 153-154: Added “where the significance of multivalency in the context of Pab1 de-mixing has been highlighted”, the key take-home message of Pab1 experiments before detailing main findings in the paper
Line 155-156: Deleted “De-mixing of Pab1 was found to only occur below both physiological ionic strength and pH (6.5-7)…” to reduce excessive experimental details
Line 157-163: Deleted “This is highlighted by the fact that all single-domain deletions still phase separate by 50°C despite having different effects on the temperature to pH ratio required for de-mixing, with the Pab1 mutant lacking the P domain (Pab1DP) displaying the most prominent effect [40]. Surprisingly, deletion of one of the RRMs (RNA recognition motif) alongside with the P-domain exhibited an even greater increase in de-mixing temperature than Pab1DP, where an almost synergistic effect can be seen” to reduce excessive experimental details
Line 154-158: Added “It was found that while the P domain of Pab1 (the proline-rich IDR) can modulate the extent of phase separation, it is not necessary for phase separation to occur. The deletion of the proline-rich domain of Pab1 showed reduced phase separation. However, the additional knockout of one of its RRMs (RNA-recognition motifs) exhibited an even greater increase in the temperature boundary required for its phase separation” to condense experiments into key points
Line 158-159: Added “This implicates that in the absence of each of its six domains, Pab1 was still able to phase separate at different temperatures, but absence of the RRM showed the most impairment to de-mix” as a simple conclusion to the primary experiment
Line 169: Deleted “specifically” – excessive filler word
Line 174: Modified “by excluding and tailoring mRNA translation” to “by using its RRMs, and to a lesser extent its P-domain, to tailor mRNA translation” for clarity
Line 186-191: Deleted “To address whether Pub1 assembly can be induced directly by a decrease in pH, cells were introduced into buffers containing a drug inhibiting proton-transport (2,4-dinitrophenol), allowing for the adjustment of intracellular pH. Pub1 foci formed in buffers at pH 5.7, while this was not the case in pH 7.5 [44]. Therefore, a change in pH alone is enough to trigger Pub1 condensate formation without fluctuation in cellular ATP levels. The protein itself forms foci in vitro, depending on the pH.” To reduce excessive experimental details
Line 210: Deleted “Initially, a significant reduction in particle mobility was measured upon energy depletion-induced acidification in the cell, where the area displaced by the tagged particle in lower pH growth media (around 5.5) was considerably less than that in pH 7.7 [47].” To reduce excessive experimental details
Line 243: Deleted “upon introduction of 1M KCl and 2M sorbitol; moreover, all OSFs formed rapidly seconds after KCl introduction and disappeared in an almost indistinguishable fashion after removal of KCl [52].” To reduce excessive experimental details
Line 244: Added “in vivo” for clarity
Line 245: Deleted “In addition to chaperones, both SG and P-body marker proteins were also tagged with GFP and observed in the same conditions, where cells were transferred to media with 1M KCl. Only the P-body proteins Dcp2 and Edc3 formed foci rapidly, whereas this was not seen for SG proteins Pub1 and Pab1 [52].” To reduce excessive experimental details
Line 243-245: Added “Highly dynamic and reversible OSFs that were found to be the chaperones Ssa1, Hsp104 and Hsp42 formed in vivo upon induction of hyperosmotic shock, which disappeared swiftly upon stress removal [55].” As brief summary of findings
Line 246-249: Deleted “Varying types of OSFs were formed in vivo in reflection of the different forms of hyperosmotic shock, such as KCl or sorbitol addition [55]. Upon introduction of KCl, small foci of the chaperones Ssa1, Hsp104 and Hsp42 were observed in vivo which were formed rapidly within seconds and disappeared after stress removal [52].” To reduce excessive experimental details
Line 251: Deleted “irreversible” as this is not always true for prions
Line 269: Added “One of the most well-researched prion-like proteins in yeast is Sup35”
Line 270: Added “…its prion form [PSI+]” to establish difference between the prion and gelation form of Sup35
Line 272-274: Deleted “The depletion of nutrient sources such as glucose leads to a change in cytosolic pH and exerts a direct effect on the phase separation of the stress-granule proteins Pab1 and Pub1” irrelevant information in Sup35 section
Line 275-283: Deleted “A study on Sup35 has demonstrated that cells only expressing the C-domain (lacking the IDR domain, NM) displayed foci in stress conditions, but remained soluble in conditions of optimal growth [5]. Thus, it appears that the GTPase catalytic domain alone, which is responsible for translation termination, is capable of forming stress-induced droplets. While the droplet formation is reversible, the speed at which these become soluble is significantly slower in the absence of the NM-domain compared to that in the wild-type strain after removal of stress [5]. Thus, the NM-domain regulates the droplet properties of Sup35 gels and probably translation termination function in stress conditions. Moreover, the reduction of translation during energy depletion correlates with Sup35 phase separation [5].” To reduce excessive experimental details
Line 275-278: Added “A study on Sup35 has demonstrated that, unlike prion formation, gelation only requires its GTPase catalytic domain which is responsible for translation termination [5]. Although this is the case, the IDR of Sup35 (NM domain) was shown to be important for governing the droplet properties of Sup35 gels [5].” As interpretation of primary experimental details
Line 282-284: Added “It is important to note that Sup35 prions are also reversible, although this often requires curing with chaperones, while gel structures appear to be reversed simply through a conditional feedback process [5, 98].”
Line 283: Added additional ref [65]
Line 286-287: Added “Sup35 is present across many fungal species [99, 100]” to introduce the species it originates from
Line 286: Added additional ref [66][67]
Line 287-290: Modified “In contrast to S. cerevisiae, Sup35 is unable to form prions in Schizosaccharomyces pombe, while both are able to form stress-induced Sup35 condensates [5]” to “In contrast to S. cerevisiae, Sup35 is unable to propagate [PSI+] in Schizosaccharomyces pombe in a Hsp104-dependent manner, while both have been shown to be able to form stress-induced Sup35 condensates [5,99]”
Line 290-291: Added “However, whether endogenous [PSI+] can be propagated in S. pombe has not yet been tested and it is possible that prion formation is mediated by other chaperone machinery.”
Line 300-301: Added “…such as nutrient deficiency” to highlight that this is one common example of stress that is known to trigger Sup35 gelation
Line 448-450: Definition of ageing modified to “Ageing is a gradual decrease in the physiology of cells with time or number of divisions” for a better and more circumspect definition.
Line During ageing, cells have difficulties to deal with accumulated aberrant protein aggregates which results partly from weakened mitochondrial activity”
Line 334-340: Added “. Snf1, a heterotrimeric protein kinase of the Snf1/AMP-activated protein kinase family, primarily ensures survival under limiting glucose conditions by activating genes involved in utilisation of sucrose, galactose or ethanol. Snf1 is also involved in cellular developmental processes such as meiosis, sporulation and ageing as well as response to other stress conditions such as oxidative stress, heat stress and regulation of various metabolic enzymes. Activation of Snf1, under glucose limitation is mediated by Std1 [79–82]” to further describe the cellular function of Std1.
Line 340: Additional references added nos.79-80, 82
Line 346-350: Added “Interestingly, Std1 when over-expressed in S. cerevisiae promotes the formation of [GAR+] prion, generating an adaptive phenotype where the cells are not reliant on glucose as primary energy source [82]. Again, just as in the case of Sup35, this demonstrates how the same protein can promote the formation of structurally and physically distinct structures depending on different environmental cues” to clarify that condensates formed by Std1 are not the same as [GAR+] prion.
Line 348: Additional reference added no. 83
Line 425: Deleted “another prion-like protein which has been shown to actually gain function when it undergoes conformational change is MAVS” to avoid repetitions.
Line 532: Reference 21 modified to “Wallace, E.W.; Kear-Scott, J.L.; Pilinpeko, E.V.; Schwartz, M.H.; Laskowski, P.R.; Rojek, A.E.; Katanski, C.D.; Riback, J.A.; Dion, M.F.; Franks, A.M.; et al. Reversible, specific, active aggregates of endogenous proteins assemble upon heat stress. Cell 2015, 162 (6),1286-98,doi: 10.1016/j.cell.2015.08.041.”